Population structure of Neisseria gonorrhoeae based on whole genome data and its relationship with antibiotic resistance

Ezewudo Matthew N. 1
Joseph Sandeep J. 1
Castillo-Ramirez Santiago 2
Dean Deborah 3 11
del Rio Carlos 1 4
Didelot Xavier 5
Dillon Jo-Anne 6
Selden Richard F. 7
Shafer William M. 8 9
Turingan Rosemary S. 7
Unemo Magnus 10
Read Timothy D. 1 tread@emory.edu
1 Department of Medicine, Division of Infectious Diseases, Emory University School of Medicine , Atlanta, GA , USA
2 Programa de Genómica Evolutiva, Centro de Ciencias Genómicas, Universidad Nacional Autónoma de México , Cuernavaca, Morelos , México
3 Children’s Hospital Oakland Research Institute , Oakland, CA , USA
4 Hubert Department of Global Health, Rollins School of Public Health of Emory University , Atlanta, GA , USA
5 Department of Infectious Disease Epidemiology, Imperial College London , London , UK
6 Department of Microbiology and Immunology, College of Medicine, Vaccine and Infectious Disease Organization International Vaccine Centre, University of Saskatchewan , Saskatoon, Saskatchewan , Canada
7 NetBio , Waltham, MA , USA
8 Department of Microbiology and Immunology, Emory University School of Medicine , Atlanta, GA , USA
9 Laboratories of Bacterial Pathogenesis, Veterans Affairs Medical Center , Decatur, GA , USA
10 WHO Collaborating Centre for Gonorrhoea and other STIs, Örebro University Hospital , Örebro , Sweden
11 University of San Francisco at California, Division of Infectious Diseases , San Francisco, CA , USA
Josenhans Christine
Electronic publication date: 2015 Mar 5
Publication date: 2015
Volume: 3
Electronic Location ID: e806
Received 2014 Nov 25; Accepted 2015 Feb 8
Copyright: © 2015 Ezewudo et al.
Copyright year: 2015
Copyright holder: Ezewudo et al.
License: This is an open access article distributed under the terms of the Creative Commons Attribution License, which permits unrestricted use, distribution, reproduction and adaptation in any medium and for any purpose provided that it is properly attributed. For attribution, the original author(s), title, publication source (PeerJ) and either DOI or URL of the article must be cited.
License URL: https://creativecommons.org/licenses/by/4.0/

Keywords: Antimicrobial resistance, Population structure, Recombination, Whole genome sequence analysis, Genetic admixture

Funding: National Institutes of Health/National Institute of Allergy and Infectious Diseases (NIAD) AI09768 R37 AI021150-29 R43 AI097688 Medical Research Service of the Department of Veterans Affairs Senior Research Career Scientist Award Centers for Disease Control and Prevention 1H25PS004311 Saskatchewan Health Research Foundation #9127 Research Alliance for the Prevention of Infectious Disease [RAPID] University of Saskatchewan Funding for this study was provided by the National Institutes of Health/National Institute of Allergy and Infectious Diseases (NIAD) grants number AI09768, R37 AI021150-29 (WMS) and R43 AI097688 (Timothy D. Read, Deborah Dean and Richard F. Selden), a VA Merit Award from the Medical Research Service of the Department of Veterans Affairs, a Senior Research Career Scientist Award from the VA Medical Research Service of the Department of Veterans Affairs (William M. Shafer) and the Centers for Disease Control and Prevention grant 1H25PS004311 (Carlos Del Rio). This work was also supported by the Saskatchewan Health Research Foundation (Grant# 9127, Research Alliance for the Prevention of Infectious Disease [RAPID]), and funding to Jo-Anne R Dillon from the University of Saskatchewan. The funders had no role in study design, data collection and analysis, decision to publish, or preparation of the manuscript.

==============================
Neisseria gonorrhoeae is the causative agent of gonorrhea, a sexually transmitted infection (STI) of major importance. As a result of antibiotic resistance, there are now limited options for treating patients. We collected draft genome sequence data and associated metadata data on 76 N. gonorrhoeae strains from around the globe and searched for known determinants of antibiotics resistance within the strains. The population structure and evolutionary forces within the pathogen population were analyzed. Our results indicated a cosmopolitan gonoccocal population mainly made up of five subgroups. The estimated ratio of recombination to mutation (r/m = 2.2) from our data set indicates an appreciable level of recombination occurring in the population. Strains with resistance phenotypes to more recent antibiotics (azithromycin and cefixime) were mostly found in two of the five population subgroups.

Introduction

Neisseria gonorrhoeae, a Gram-negative bacterium, causes gonorrhea, the most common bacterial sexually transmitted infection (STIs), causing more than 106 million cases per year globally (World Health Organization (WHO), 2012). The only effective option for treating the disease and stopping its spread has been the use of antimicrobial therapy. Currently, there is no vaccine to prevent infection. Antimicrobial treatment options have diminished over time due to the progressive emergence of antimicrobial resistance (AMR) to drugs previously used to treat gonorrhea, and the paucity in the development of newer antibiotics that could effectively eradicate the pathogen (Ohnishi et al., 2011; Unemo & Shafer, 2014).

AMR evolution should be considered in the context of the genetic structure of the N. gonorrhoeae population. Early work by O’Rourke & Stevens (1993) using electrophoretic analysis of enzymes of the pathogen and serological methods suggested that gonococci have a non-clonal sexual or panmictic population structure. More recent studies have also suggested high rates of recombination within the Neisseria genus (Didelot & Maiden, 2010). High levels of recombination could confound studies of the gonococcal populations, especially if the studies are based on few genetic loci within strains as compared to the entire genomes. Recent multi-genome studies have focused on either a restricted geographic region (Vidovic et al., 2014) (genomes also included in present studies or on a small subset of the N. gonorrhoeae population (Grad et al., 2014)). Hence, there is a need for studies aimed at understanding the global N. gonorrhoeae population structure at the whole genome scale.

Past AMR studies using limited numbers of gonococcal strains from specific geographic regions of the globe have mostly focused on a number of representative genes or genetic regions of the genome to elucidate underlying mechanisms of antibiotic resistance (Hagman & Shafer, 1995; Lindberg et al., 2007; Ohneck et al., 2011; Thakur et al., 2014; Tomberg et al., 2013; Unemo & Shafer, 2014; Unemo, Golparian & Hellmark, 2014; Unemo, Golparian & Nicholas, 2012; Zhao et al., 2009). Extensive genome sequencing studies have yet to be conducted on a diverse collection of strains from different geographical locations and collected over longer time periods. Our approach in this study builds on recent multi-genome studies (Grad et al., 2014; Vidovic et al., 2014), with the goal of using whole genome analysis to elucidate two processes: (1) the population structure and dynamics of Neisseria gonorrhoeae and (2) the correlation between this population differentiation and AMR evolution in gonococci. Our genome analysis of strains from multiple sites across the world offers a geographic diversity of N. gonorrhoeae isolates, providing more depth in genome-wide studies of this pathogen and identifying possible sub-populations impacting AMR and evolution within the species.

Materials and Methods

Neisseria gonorrhoeae isolates

Sixty-one N. gonorrhoeae isolates of diverse origin were obtained. These included isolates from the Gonococcal Isolation Surveillance Program (GISP) site covering Atlanta, Miami, New York city and North Carolina in the United States (n = 21), from Canada (primarily, Saskatchewan) and Chile (n = 1) (Vidovic et al., 2014) (n = 24), and from WHO global collaborations; Sweden (n = 7), Norway (n = 3), Japan (n = 2), Austria (n = 1), Pakistan (n = 1), Philippines (n = 1), and Australia (n = 1). Phenotypic determination of the minimum inhibitory concenterations (MICs) of all isolates was performed using the agar dilution method or the Etest method (bioMerieux), according to the instructions from the manufacturer. The strains sequenced in this study were tested for resistance to primarily three antibiotics, tetracycline, azithromycin and cefixime, with breakpoints for resistance set at 2, 2.0, and 0.25 µg/mL, respectively, based on the CDC MIC (minimum inhibitory concentration) breakpoints for testing in the GISP protocol (http://www.cdc.gov/std/gisp/gisp-protocol07-15-2010.pdf). Antibiotic resistance profiles of the Canadian strains have been previously reported (Vidovic et al., 2014). Details of the different isolates with their NCBI accession numbers are presented in Table 1.

Table 1 Location and date of collection of the N. gonorrhoeae strains including Sequence Types and MICs of the different strains to the antibiotics azithromycin, cefixime and tetracycline.

The MIC breakpoint value for azithromycin resistance is 2 µg/mL, for cefixime 0.25 µg/mL, and for tetracycline 2 µg/mL, based on the CDC breakpoints for antibiotic testing.

Strain name	Location	Date	MLST	Azithromycin
(MIC)	Cefixime
(MIC)	Tetracycline
(MIC)	
CH811	Chile	1982	1583	0.25	0.008	2	
GC1-182	Canada	1982	1583	0.5	0.008	4	
SK708	Canada	2006	1594	1	0.016	0.5	
SK1902	Canada	2006	10935	0.25	0.002	256	
SK6987	Canada	2006	10010	1	0.016	4	
SK7461	Canada	2008	1901	0.5	0.032	8	
SK7842	Canada	2006	10010	1	0.016	8	
SK8976	Canada	2006	1594	0.06	0.004	2	
SK12684	Canada	2006	31129	0.5	0.016	8	
SK33414	Canada	2007	1928	0.25	0.008	4	
SK14515	Canada	2005	1893	0.25	0.016	2	
SK15454	Canada	2007	1585	0.06	0.004	2	
SK16259	Canada	2007	1893	0.125	0.008	4	
SK16942	Canada	2005	1893	0.125	0.016	2	
SK17973	Canada	2006	1893	1	0.016	8	
SK22871	Canada	2007	8122	0.125	0.004	4	
SK23020	Canada	2006	1901	0.25	0.125	16	
SK28355	Canada	2007	1893	0.25	0.016	4	
SK29344	Canada	2007	10010	0.125	0.008	4	
SK29471	Canada	2005	1893	0.25	0.016	2	
SK32402	Canada	2007	8153	0.5	0.016	4	
SK36809	Canada	2007	8126	2	0.008	8	
SK39420	Canada	2008	1585	0.5	0.016	0.5	
ALB0303	USA	2011	1588	0.03	0.015	16	
ALB0403	USA	2011	1901	1	0.125	4	
ATL0103	USA	2011	10931	0.5	0.015	0.25	
ALB0102	USA	2011	1901	0.25	0.06	2	
ATL0105	USA	2011	1588	0.06	0.015	0.25	
ATL0108	USA	2011	1584	0.03	0.015	0.25	
ATL0117	USA	2011	10932	0.125	0.015	16	
ATL0121	USA	2011	1902	0.5	0.03	1	
ATL0125	USA	2011	1901	0.25	0.015	1	
ATL0508	USA	2011	1585	0.06	0.015	16	
ATL0513	USA	2011	1893	0.25	0.03	2	
MIA0202	USA	2011	1901	0.5	0.03	2	
MIA0309	USA	2011	1931	0.125	0.015	16	
MIA0310	USA	2011	1584	0.03	0.015	16	
MIA0510	USA	2011	1901	1	0.03	2	
MIA0515	USA	2011	1901	0.25	0.03	16	
MIA0516	USA	2011	1901	0.5	0.06	8	
NOR0306	USA	2011	1583	0.25	0.015	2	
NYC0507	USA	2011	1901	0.25	0.06	2	
NYC0513	USA	2011	1901	0.25	0.06	4	
MUNG1	Canada	1991	10934	0.125	<0.016	0.25	
MUNG3	Japan	2003	7363	0.25	0.5	2	
MUNG4	Japan	1996	1590	0.5	0.25	4	
MUNG5	Philippines	1992	1901	0.25	<0.016	1	
MUNG6	Australia	2001	10008	0.125	<0.016	16	
MUNG8	USA	2001	8127	2	<0.016	0.5	
MUNG9	Sweden	2010	1901	0.5	1	2	
MUNG12	Norway	2010	1901	0.5	0.25	4	
MUNG14	Norway	2010	1901	0.5	0.25	4	
MUNG15	Austria	2011	1901	0.25	1	2	
MUNG17	Sweden	2010	1892	1	0.5	2	
MUNG18	Norway	2010	10933	0.125	<0.016	2	
MUNG19	Sweden	2010	1580	>256	<0.016	2	
MUNG20	Sweden	2013	7363	0.25	0.5	2	
MUNG21	Pakistan	2008	1902	1	0.032	2	
MUNG23	Sweden	1998	1585	0.064	<0.016	0.125	
MUNG25	Sweden	1998	1901	0.125	<0.016	0.5	
MUNG26	Sweden	1999	1584	0.064	<0.016	0.5	

Sequence generation and assembly

The N. gonorrhoeae strains were shotgun (WGS) sequenced using the Illumina HiSeq™ instrument (Illumina, San Diego, California, USA), utilizing libraries prepared from 5 µg of genomic DNA for each sample. The average sequencing coverage was 225. The sequencing reads were filtered using the prinseq-lite algorithm (Schmieder & Edwards, 2011) to ensure only sequence reads with average phred score ≥30 were used. The reads for each project were then assembled de novo, using the velvet assembler program (Zerbino & Birney, 2008). The optimal kmer length for each assembly prior to assembly was determined using the velvet optimizer algorithm (Gladman & Seemann, 2012). Data was deposited in the NCBI Sequence Read Archive public database (Accession # SRA099559) (Table 1). For this study, we included an additional 14 draft genome sequences of N. gonorrhoeae strains, downloaded from the NCBI draft genomes database (NCBI Bioproject numbers: PRJNA55649, PRJNA55651, PRJNA55653, PRJNA55905, PRJNA46993, PRJNA55657, PRJNA55655, PRJNA55659, PRJNA55661, PRJNA55663, PRJNA55665, PRJNA55667, PRJNA55669, PRJNA55671, and the reference genome sequence Ref_FA_1090 (NC_002946.2).

Genome-wide phylogeny and pangenome analysis

The assembled genomes were annotated individually using the NCBI PGAP annotation pipeline to give predicted proteome for each of the strains. The orthologs were determined by OrthoMCL (Li, Stoeckert & Roos, 2003), which uses bi-directional BLASTP scores of all the protein sequences to perform Markov clustering in order to improve sensitivity and specificity of the orthologs. For the OrthoMCL analysis, we used a BLASTP E-value cut-off of 1e-05, and inflation Markov clustering parameter of 1.5. Core genes were defined as the orthologous genes that are shared among all the N. gonorrhoeae strains used in this analysis.

The nucleotide sequences of all the core genes were concatenated together and core-gene nucleotide alignment was conducted using progressive MAUVE (Darling et al., 2004). Similarly, whole genome amino-acid alignment was also generated by concatenating the deduced amino-acid sequences of all the core genes generated using MUSCLE (Edgar, 2004), and to form a super protein alignment. Homoplasious sites were removed from the whole-genome nucleotide alignment using the Noisy software (Dress et al., 2008). The protein alignments were filtered by GBLOCKS (Talavera & Castresana, 2007) using default settings to remove regions that contained gaps or were highly diverged. A maximum likelihood (ML) tree from the same data set was created using the GTR and JTT substitution models for the nucleotide and protein alignment respectively and the GAMMA evolutionary model (Stamatakis, 2014). The majority rule-consensus tree was generated from 200 bootstrap replicates of the model. Linear regression of the root-to-tip distances against the year of isolation was performed using the Path-O-Gen tool (http://tree.bio.ed.ac.uk/software/pathogen/).

Multi-locus sequence typing (MLST) locus analysis

MLST is a genotyping tool for Neisseria based on sequencing of 7 core housekeeping genes (Jolley & Maiden, 2010). There are currently close to 11,000 individual Neisseria sequence profiles in the publicly available MLST database (http://pubmlst.org). We utilized a custom python script mlstBLAST.py (http://sourceforge.net/projects/srst/files/mlstBLAST/) to perform a BLAST search of these genes across all the strains in our data set and identified the sequence type (ST) for each strain. Novel alleles of the locus and STs were submitted to the MLST database. A phylogeny of the concatenated DNA sequences of all the N. gonorrhoeae STs in the MLST public database was created using the neighbor joining distance matrix approach of the PHYLIP (Felstein, 1989). Mean nucleotide distance for the sequence alignments and MLST genes was computed using MEGA software (Tamura et al., 2013).

Estimating population parameters and homologous recombination

ClonalFrame (Didelot & Falush, 2007) utilizes a statistical framework to reconstruct the clonal genealogy as well as identify the regions along the genomes that has been affected both by recombination and mutation. The model uses a Bayesian approach to predict the phylogenetic relationship in the sample set, given the whole genome sequence alignment data. The input genome alignment data was the core genes (n = 1,189) alignment generated from MAUVE. Four independent ClonalFrame runs were performed for 40,000 iterations, with the first 20,000 discarded as burn-in. This allowed the model parameters to converge, and each of the 4 runs were checked for the consistency of the estimated parameters as well as the consistency of the topology of the inferred clonal genealogies.

Population structure analysis

The program BAPS (Bayesian Analysis of Population Structure) version 5.3 (Corander & Marttinen, 2006; Tang et al., 2009) was used to infer the underlying population structure of the 76 N. gonorrhoeae strains in the sample set. SNPs from the core MAUVE alignment, with gaps removed were converted to a BAPS input file, which is a representation of all the polymorphic loci in the multi-sequence alignment. BAPS applied a Bayesian model to predict the likelihood of a population structure, given the input data and non-parametric assumption approach to trace ancestry of the different individuals in the sample set. For the mixture analysis we used the ‘Clustering of individuals’ approach. We ran a preliminary analysis to evaluate the approximate number of genetically differentiated groups using a vector from 2 to 40 K values, where K is the maximum number of groups. Given that 5 groups was the K value with the best log likelihood, we ran a second analysis using from 3 to 7 K values and again the best K value was 5 groups. We used the ‘Admixture based on mixture clustering’ module for the admixture analysis. For the analysis, the minimum population and the admixture coefficient for the individuals was then set to 5. For the reference individuals from each population and the admixture coefficient for reference individuals we used the values as described by Castillo-Ramírez et al. (2012). In addition, population structure analysis of the sample set using the fineSTRUCTURE tool (Lawson et al., 2012) was performed. The fineSTRUCTURE analysis was a two step process-(1) ChromoPainter algorithm was used to generate the co-ancestry matrix from the genome-wide haplotype data using the linkage model. (2) The fineSTRUCTURE algorithm performed a model-based clustering using a Bayesian MCMC approach to predict the likelihood of a population structure given the input data and non-parametric assumption approach to trace ancestry of the different individuals in the sample set. The fineSTRUCTURE approach was used to corroborate the findings from the BAPS population structure analysis.

Mapping the movement of DNA between Neisseria gonorrhoeae clades

We traced the flow of recombination between strains into five different subgroups in the phylogeny determined from the subgroups of the population defined by the BAPS analysis. We created a BLAST database of the whole genome sequence of all 76 strains in the sample set and included 14 whole genome sequences of all other Neisseria species (NC_008767.1, NC_014752.1, NC_017512.1, NC_017516.1, NC_003112.1, NC_010120.1 NC_017501.1, NC_017514.1, NC_017517.1, NC_003116.1, NC_013016.1, NC_017505.1, NC_017515.1, NC_017518.1) that are present in the NCBI database. Next, we performed a BLASTN search for each of the genomic region within the strains identified by ClonalFrame to be under recombination, selecting the best hit within the sequences in the database we created, with an identity of >98%, to be the source of the recombined region. We also removed BLAST matches found in strains from similar subgroups as the source of the recombined region. We used the migest package (http://cran.r-project.org/web/packages/migest/) implemented in the R statistical language to create a circular representation of the matrix of relationship between the subpopulations identified by BAPS based on the purported recombination between strains in the different subgroups. We also supplied migest with the matrix from BAPS admixture analysis and recreated the circular flow of recombination across only the subpopulations as defined by BAPS.

Comparison of nucleotide substitution rates

Amino acid sequences were aligned using MUSCLE sequence aligner (Edgar, 2004). The amino acid sequence alignment was converted to nucleotide alignment based on the corresponding gene sequence using PAL2NAL (Suyama, Torrents & Bork, 2006) and we implemented the YN00 method of the PAML package (Yang, 2007) to calculate the pairwise dN/dS ratios for the strains (Rocha et al., 2006). The contribution of each strain to the overall variation in the dN/dS rates across the sample set was estimated using ANOVA (Analysis of Variance) approaches.

Analysis of positive selection

For the analysis of positive selection within core genes of the strains in the sample set, we first identified and removed core genes that have signals of homologous recombination using three methods of Pairwise Homoplasy Index (PHI), Neighbor Similarity Score (NSS) and the maximum χ2 method. The three methods are implemented in the PhiPack package (Sawyer, 1989). A window size of 50 nucleotides was used to run the methods in the package, and genes shown to have significant probability of homologous recombination by a majority of the methods were not used for the positive selection analysis. Next, we identified core genes under positive selection using codeml of PAML tool version 4.7 (Yang, 2007). We applied the branch-sites test for positive selection Model A test 2 of the tool, to identify genes under positive selection population groups. For each of the clades, we performed the Likelihood Ratio Test (LRT) for two hypotheses—the null hypothesis is the existence of neutral selection as implemented in the null model versus the alternative hypothesis implemented in the test model for positive selection. The LRT was performed to a degree of freedom of 1, and we corrected for multiple testing using the False discovery rate approach (FDR) (Benjamini & Hochberg, 1995). We further identified the Gene Ontology (GO) terms and functional characterizations of the genes under positive selection (see Table 2) and performed an enrichment test for functionality of these genes using the blast2go test pipeline (Götz et al., 2011).

Table 2 Core genes of N. gonorrhoeae under positive selection in the different clades of the phylogeny of strains in the sample set.

Gene	Clades present	Gene ID (reference genomeFA 1090)	
PorB	1, 3, 5	NGO1812	
Acetate kinase 2	5	NGO1521	
Primosomal replication protein	3	NGO0582	
DNA Helicase	3, 5	NGO1196	
Hypothetical protein	5	NGO0880	
Hypothetical protein	5	NGO1847	
Hypothetical protein	5	NGO1948	
ComA	5	NGO0276	
Chaperone protein HscA	5	NGO0829	
tRNA-ribosyltransferase	5	NGO0294	
RNA polymerase Subunit β	5	NGO1850	
ArsR family transcriptional regulator	5	NGO1562	
Hypothetical protein	5	NGO0165	
PriB	5	NGO0582	
ABC transporter subunit	3	NGO2088	
Hypothetical protein	3	NGO1984	
tRNA pseudouridine synthase B	3	NGO0642	
Prolyl endopeptidase	1	NGO0026	
Apo-lipoprotein N-acyltransferase	1	NGO0289	
Sodium dependent transporter	1	NGO2096	
Phage associated protein	1	NGO1012	
Hypothetical protein	4	NGO0914	

Confirming known predictors of antibiotic resistance phenotype

We downloaded from NCBI reference DNA sequences of resistance determinants that have been shown in the literature to underlie the resistance phenotype we have observed in our sample set (see Table 3), and performed a BLASTN search for each of these DNA sequence regions across all the strains in the database of whole genome sequences. For convenience, the contigs for each assembly were ordered into one pseudocontig after tiling to the reference genome FA1090, using the ABACUS tool (http://abacas.sourceforge.net/).

Table 3 Known antibiotic resistance determinants in sample set.

The description includes the PubMed reference ID and associated resistance phenotypes of these determinants in N. gonorrhoeae. Cephalosporin antibiotics include cefixime and ceftriaxone, while macrolides include erythromycin and azithromycin.

Gene name	FA1090 Reference
locus_tag/
Gene Bank ID	Genetic Mutations	Resistance Phenotype	References	
mtrR	NGO1366	G45D, A39T (glycine and aspartate substitutions)	Decreased susceptibility to macrolides and beta-lactams	PMID: 18761689	
mtrCDE promoter	NGO1366	Single nucleotide deletion on reference genome position 1327932	Decreased susceptibility to macrolides and beta-lactams	PMID: 18761689	
penB	NGO1812	G101K, A102D (glycine and alanine substitutions)	Decreased susceptibility to third-generation cephalosporins	PMID: 17420216	
Mosaic penA	NGO1542	Mosaic pattern amino acid substitutions form position 294 to end of gene	Decreased susceptibility to third-generation cephalosporins	PMID: 20028823	
rpsJ	NGO1841	V57M	Decreased susceptibility to tetracycline	PMID:16189114	
23S rRNA	AF450080	C2611T (Cystine to Threonine substitution)	Decreased susceptibility to Azithromycin	PMID: 12183262	
tetM	N/A	Horizontally transferred determinant on plasmids	Resistance to tetracycline(MIC > = 16 µg/ml)	PMID: 21349987	

We selected the top hit (with identity match of 98% or more) for each sequence (strain) in the database and parsed the alignment between the query and the subject sequence in the database for the presence or absence of the underlying resistance genetic mutations as suggested in the literature.

Results and Discussion

Genome-wide homologous recombination in diverseN. gonorrhoeae

We sequenced 61 recent clinical isolates primarily from the US and Canada but also single representatives from other countries, including Japan, Pakistan, Australia, Austria, Philippines, Norway and Sweden using the WGS approach to a high average coverage (average 225-fold read redundancy). De novo assemblies based on these data produced a set of contigs that represented draft, unordered representations of the genomes with high sequence quality. A preliminary phylogeographical analysis of the Canadian isolates (n = 23) was recently published (Vidovic et al., 2014). For the analysis, we included the 14 N. gonorrhoeae draft NCBI genome sequences (12 from the US and 2 from Europe) and the genome sequence of the FA1090 N. gonorrhoeae reference strain. The 76 were assigned into 23 previously described MLST STs and four new STs (10931,10932,10933,10934). The genetic diversity (measured as pairwise nucleotide distances of MLST loci) of the strains in this study was about half that of the N. gonorrhoeae strains as a whole (0.001 substitutions per site in our study compared to 0.002 in the large MLST set), and the STs from our sample set were represented across the different clades of a phylogeny of housekeeping genes of N. gonorrhoeae strains found in the MLST database (see Fig. S5). Alignment of the shotgun assembly to reference genome FA1090 (NC_002946.2), yielded 10,962 SNPs in the core region (conserved in all strains). The average per nucleotide diversity in the core genome regions was 0.003.

Homologous recombination is known to play a role in shaping bacteria populations (Didelot & Maiden, 2010). The ClonalFrame tool (Didelot & Falush, 2007) detected 952 independent recombination events, covering more than 50% of the reference genome. The average size of the recombination regions identified was 360 base pairs. The estimate for the ratio of effects of recombination and mutation (r/m) for our strain set was 2.2, a relatively high value for bacterial species (Didelot & Maiden, 2010) (Fig. S3) and quite similar to the r/m estimate of 1.9 based on the whole genome alignment on a less genetically diverse group of N. gonorrhoeae strains reported by Grad et al. (2014).

We constructed a maximum likelihood phylogeny of the core genome of the 76 strains using the RAxML program (excluding regions identified as potentially recombinant) (Fig. 1). This tree had similar topology to the clonal frame that determined by the eponymous software (Fig. S3). The tree showed multiple clades but the there was no strong signal of genetic isolation by distance at the continental scale. The rate of the molecular clock was estimated to be 8.93 × 10−6 mutations per year based on the slope of the regression of the root-to-tip divergence with isolation dates (see Fig. S2). This value was similar to those obtained in other bacterial studies, ranging from 8.6 × 10−9 to 2.5 × 10−5 (Zhou et al., 2013). However, because the temporal signal was weak in the root-to-tip analysis (Fig. S2), we did not use these data for Bayesian phylogeny analysis using the BEAST phylogeny tool (Drummond & Rambaut, 2007).

Figure 1 Phylogeny of N. gonorrhoeae strains in the sample set.

(A) (Top). Maximum Likelihood phylogeny of sequenced strains of N. gonorrhoeae Branches with boostrap value >80% for branches are indicated. Taxa are highlighted based on 5 different subgroups defined by BAPS. Annotations next to the leaves are colored based on location of isolation; Canada is colored red, US blue, Europe green, Asia purple and the lone strains from Australia (MUNG6) and Chile (CH811) is colored brown. (B) Unrooted phylogeny based on the same tree.

Neisseria gonorrhoeae population structure and biogeography

Given that recombination was frequent in these genomes, we sought to evaluate the genetic substructure of the population. We used two complementary methods. BAPS (Tang et al., 2009) predicts the likelihood of a population structure given the input data and uses a non-parametric assumption approach to trace ancestry. fineSTRUCTURE (Lawson et al., 2012), on the other hand, uses similar methods of predicting population substructure, but to a finer detail and does not assume a prior optimum number of subpopulations (K). The BAPS tool identified 5 subgroups within the N. gonorrhoeae population from the strains within the sample set (Fig. 3). As expected, members with the same subgroup ancestry generally were found near each other when mapped on the ML phylogeny constructed using the nonrecombining portion of the genome. On the other hand, fineSTRUCTURE identified 30 genetic subgroups within our sample set (Fig. S4). However, individual members of each of the fineSTRUCTURE subgroups belong to the same BAPS subgroup. Each of the five BAPS subgroups contained strains from multiple continents based on geography or location of isolation (Fig. 1). It was particularly interesting that each BAPS cluster had at least one US strain and one Canadian strain. The BAPS analysis revealed a complex relationship between Group 3 and 5, with the latter group 5 separated into two group 3 clades (Fig. 1). Group 3 strains in clades closely related to group 5 showed significant genetic import from group 5. It is possible that the extent of admixture occurring in group 3 and 5 may have caused misidentification. Also, strains of sequence type (ST) 1901, which is the most abundant ST in our sample set all belong to subgroup 1, hinting at a correlation between BAPS subgrouping and MLST.

We assessed patterns of genetic drift effects in the population by estimating the pairwise substitution rates between all the core gene orthologs for the strains and determining the mean dN/dS ratio for each strain. The mean pairwise dN/dS ratios for each strain are shown in Fig. 4. There was significant variation in the mean dN/dS ratios among the strains (ANOVA p-value = 2.0e-16). The overall mean of the dN/dS estimate was 0.3184, similar to the 0.402 value estimated for the bacterial pathogen Chlamydia trachomatis (Joseph et al., 2012). The mean dN/dS ratio for the five subgroups respectively was (0.32412976, 0.33325164, 0.31273103, 0.30952504 and 0.30990092). There is no significant difference between group mean dN/dS ratios (p-value =0.921, t test of means).

The mean dN/dS ratio for strains from the Canadian region was 0.3279, which was above the overall mean ratio, while that for strains collected in the US was 0.31708, which is below the overall pairwise dN/dS mean ratio for the sample set. This was also a statistically significant difference (p-value 0.0018 for t test of means), suggesting a possible geographical effect within this subset of strains.

Genetic admixture within N. gonorrhoeae and with other Neisseria species

In order to understand the flow of genetic information between the strains from five different subgroups defined by the BAPS analysis (Fig. 3) as well as strains from other Neisseria species, we used two independent approaches. The first was to search each of the 952 recombination regions identified by ClonalFrame for a best BLASTN match from another subgroup or Neisseria species (We created a blast database of the 76 genomes from this study and representative strains from the Neisseria genus.) (Fig. S3). In parallel, we also counted the occurrence of co-ancestry of genetic markers revealed by the BAPS analysis. Both the BAPS and BLAST analyses suggested that group 3 was the most common nexus of homologous recombination between other clades, consistent with its basal phylogenetic status. In the BAPS-based network groups 1 and 2, and to a greater extent, group 5, were primarily DNA donors to group 3 (Fig. 2B). But this pattern was less visible in the BLAST network (Fig. 2A). It is notable that more than 90% of the recombination with strains from other Neisseria species occured in groups 2 and 3. Group 5 stood out as a significant source of genetic exchange into strains in group 3.

Figure 2 Pathways for exchange of genetic materials between populations.

(A) Recombination pattern traced from BLAST results of similarity of recombined regions between the subgroups defined by BAPS of N. gonorrhoeae. The clade showed as external represents strains from other Neisseria species. (B) Exchange of genetic materials among subgroups within the sample set as defined by BAPS admixture analysis. Colored base sub-sectors of the circle for each subgroup in the diagram represents outflow of genetic material while blank or white colored sub-sectors represent inflow of genetic materials to the subgroups. The figure was created using the migest package of the R statististical tool.

Figure 3 Population subgroups from strains of N. gonorrhoeae in the sample set defined by BAPS.

The names for each strain in the different subgroup are indicated at the bottom of the plot on the x-axis, while the fineSTRUCTURE group labels for each strain is indicated on top of the plot. This figure shows that the structure of the Neisseria gonorrhoeae population divides into five subgroups. Each color represents one of the genetically differentiated groups and each vertical colored bar corresponds to one isolate. When the vertical bars show two colors, each color corresponds to one of the groups and this is evidence for admixture; the proportion of every color in the bar reflects the proportion of the genome coming from the group represented by that particular color.

Figure 4 Boxplot of mean dN/dS ratio pair-wise comparison of core genes of each of the strains of N. gonorrhoeae in the sample set.

The box plot is colored by subgroups within the Neisseria population, defined by the BAPS tool.

The genetic relatedness of the strains in the sample set or the purported sharing of genetic materials across the different subgroups shown by the BAPS figure paralleled the pattern revealed by the BLAST clonal frame analysis (p-value = 0.048, Mantel test for comparing the distance matrices of the five populations between both methods). The exchange of genetic materials from other Neisseria species was not accounted for in the BAPS admixture analysis. Based on the BLAST analysis, the proportion of DNA transferred within N. gonorrhoeae compared to arriving from Neisseria strains the species was 729 out of 849 intra-specific genetic events. This finding is line with the “fuzzy species” concept of Fraser, Hanage & Spratt (2007): while N. gonorrhoeae is not sexually isolated, DNA flow seemed predominantly through intra-specific exchanges.

Genes under positive selection

Of the 1,189 core genes, we identified 352 genes as likely to contain past recombination histories using the PHIPACK tests (Sawyer, 1989). Thirty-one genes within the subset of 837 non-recombining core genes were found to be under positive selection using the tests implemented by the PAML software (‘Materials and Methods’). BAPS subgroup 5 had the highest number of core genes under selection (14) followed by subgroup 3 (7). While we found no significant enrichment of genes under positive selection in any of the functional classes in the Gene Ontology (GO) database, the functions of the best match proteins from genes under positive selection can be broadly classified to genes involved in DNA or RNA synthesis of gene expression, membrane or transport proteins, and, to a lesser extent, genes involved in metabolic pathways in the bacterial cell (Data S2 spreadsheet). Of the 352 genes found to have signals of recombination, we found no significant enrichment of the genes in any of the functional classes in the GO database. The functions of these genes could broadly be classified into 2 groups: genes encoding membrane and transport proteins; and those involved in metabolic pathways in the cell (Data S3).

In regard to antibiotic resistance and selection, the most interesting gene found to be under positive selection was porB (Smith, Maynard Smith & Spratt, 1995), which has been shown to be involved in mechanisms of resistance to penicillins, macrolides, cephalosporins and tetracyclines (Unemo & Shafer, 2014). porB exhibited signals of selection in subgroups 1, 3 and 5—the groups that harbored most of the antibiotic resistant strains in our sample set (Fig. 5). comA, which encodes a membrane protein necessary for competence of N. gonorrhoeae, was also found to be under selection in a handful of strains that make up subgroup 5. This finding is of interest in regards to the potential for DNA uptake in these strains, since they appear to be primarily DNA donors, rather than recipients in genetic exchanges (Fig. 2). Other genes putatively under selection included a stress response gene, a gene encoding a chaperone protein of the HscA family and a number of proteins: ribosyl transferase, RNA polymerase, and an arsR family transcriptional regulator, which were all linked to gene expression. Genes under positive selection in subgroup 3 were also mainly involved in gene expression or DNA metabolism, including DNA helicase and tRNA pseudo uridine synthase. Most of the genes with known functions, identified to be under positive selection in subgroup 1 were either membrane-associated or transport proteins.

Figure 5 Representation of antibiotic resistance profile of N. gonorrhoeae strains across different subgroups of the population.

The topology is identical to the ML tree in Fig. 1.

Analysis of known genetic predictors for AMR phenotypes

A substantial amount of research effort over the past 10 years has been devoted to understanding the genetic basis of drug resistance in N. gonorrhoeae (Garvin et al., 2008; Unemo & Shafer, 2011; Veal, Nicholas & Shafer, 2002; World Health Organization (WHO), 2012). Since there is a an increasing interest in the direct attribution of resistance phenotypes based on genome sequencing, we attempted to ascertain how knowledge of existing variants could be applied to the N. gonorrhoeae genomes in this study. We searched for variants known to underlie resistance to 3 antibiotics classes within our study (Table 3). In terms of subgroup distribution, tetracycline resistance was found in each of the 5 population subgroups, azithromycin resistance was present in only 2 of the strains tested (SK36809 and MUNG8) and restricted to subgroup 2 (Fig. 5), and cefixime resistance was found in subgroup 1 and subgroup 2. We identified genes responsible for resistance to the drugs tested in this works using literature searches and the CARD antimicrobial resistance database (McArthur et al., 2013).

The tetM resistant determinant, which confers high-level resistance to tetracycline, is borne on plasmids and is transferred either through conjugation or transformation (Knapp et al., 1988; Morse et al., 1986; Turner, Gough & Leeming, 1999). It was found in only 5 of the 10 strains with high-level resistance to tetracyline (MIC equal or greater than 16 µg/ml). Strain SK1902, one of the 5 strains with the tetM determinant, had a significantly higher MIC (>256 µg/ml) than the rest (see attached Data S3). Other strains with reduced susceptibility or chrosomally-mediated resistance to tetracycline, i.e., without the tetM determinant do have other corresponding chromosomal mutations on one or more of the resistance loci: mtrR (including its promoter), penB, rpsJ. Only one strain (ATL0508) within the sample set exhibits resistance to tetracycline in the laboratory, without the presence of any of the known resistance determinants of the tetracycline resistance phenotype.

Different “mosaic” penA alleles are thought to have developed from recombination with portions of DNA transferred horizontally from commensal Neisseria and/or N. meningitidis and underly decreased susceptibility or resistance to cephalosporins by preventing their binding action on the encoded mosaic PBP2 (Ameyama et al., 2002). The mosaic penA XXXIV (Ohnishi et al., 2011; Grad et al., 2014; Unemo & Shafer, 2014) had the best positive predictive value of all the known resistance determinants we searched for within our dataset, being present in 6/7 of the strains resistant to cefixime. This result echoed the observations made by Grad et al. (2014) in their epidemiologic study of N. gonorrhoeae strains. The other loci (i.e., mutations in the mtrR, mtrCDE operon promoter region and penB gene) also proven to enhance the MICs of cephalosporins (Unemo & Shafer, 2011; Warner, Shafer & Jerse, 2008) did not have a similar predictive property within strains in our data set. These variants were seen in 2 out of 7 and 3 out of 7 cefixime resistant strains, respectively. MUNG17 is the only strain in the sample set that has an elevated MIC (0.38 µg/mL) to cefixime that we could not find any of the known resistance determinants within its genome sequence (see attached Data S1).

Resistance to azithromycin can be mediated by mutations in the previously mentioned penB and mtr operon genes as well as mutations found in the 4 different alleles of the 23S rRNA gene that inhibits protein synthesis (Chisholm et al., 2009; Palmer, Young & Winter, 2008; Starnino, Stefanelli & Neisseria gonorrhoeae Italian Study Group, 2009). The 23S rRNA mutation allele was found in one (SK36809) of the two strains with the azithromycin resistance phenotype. The other azithromycin resistant strain, MUNG8, did not have the 23S rRNA resistance determinant or any of the other known mutations in the mtrR or penB loci (see attached Data S1).

Conclusions

Our study suggested that N. gonorrhoeae globally is made up of at least five genetic subpopulations. That individual strains from the subpopulations are from diverse geographical locations confirms the cosmopolitan nature of the pathogen. This suggested a population structure with multiple waves of rapid international expansion. Subgroup 3 strains may be the nexus for gene exchange within the species. Groups 1 and 2 might be the most recently branched and contain a higher proportion of resistant isolates to more currently used antibiotics. Given the importance of the antibiotic resistant phenotype, these may be emerging lineages that are expanding within N. gonorrhoeae. It will require a more extensive study with a broader number of strains to ascertain this suggested evolutionary trend. Our analysis confirms earlier studies that showed an appreciable effect of recombination within the population. This could be playing a role in the evolution of AMR in the bacterium, as strains with resistance phenotypes to currently used antibiotics are mostly within similar population sub-groupings.

Although most of the known predictors that underlie the observed resistance phenotypes were accounted for in the strains we studied, they could not explain some of the phenotypes of several strains. These findings suggested that a broader genome search of a large number of whole genomes from strains of this pathogen could yield candidate novel variants that may explain some of the “missing” antibiotic resistance phenotypes we have observed.

In general, large genome sequencing studies examining a high number of temporally and geographically diverse N. gonorrhoeae isolates are essential to elucidate the evolution and diversity of the N. gonorrhoeae species as well as associations between genomic content, antibiotic resistance and clinical outcome of treatment.

Supplemental Information

Supplemental Information 1 Supplemental Information

Click here for additional data file.

Data S1 Meta data of Strains in sample set

Click here for additional data file.

Data S2 Gene Ontology (GO) and functions for core genes in sample set, under positive selection

Click here for additional data file.

Data S3 Gene Ontology (GO) and functions of core-genes influenced by recombination in the sample set

Click here for additional data file.

Figure S1 Representation of the pan-genome of strains of N. gonorrhoeae in the sample set

Each bar is a count of the number of genes (technically gene clusters) found in n genomes (n = 76). Area to the right of the red line represents the extended core genes; to the left are the non-core genes.

Click here for additional data file.

Figure S2 Root-to-tip plot as a measure of molecular rate of change in strains of N. gonorrhoeae species over time produced using the Path-oGen software

Click here for additional data file.

Figure S3 Heat map of pairwise recombination between strains of N. gonorrhoeae in the sample set

Putative recombination hotspots are on positions 8, 11 and 16 MB on the x-axis of the plot.

Click here for additional data file.

Figure S4 Heat map of N. gonorrhoeae population structure

The figure was generated using fineSTRUCTURE tool, representing pairwise genetic relationship between the strains in the sample set.

Click here for additional data file.

Figure S5 Phylogenetic representation of N. gonorrhoeae individual Sequence Types obtained from the MLST public database (http://pubmlst.org)

The blue colored spots represent majority of the sequence types present in our sample set.

Click here for additional data file.

Genome sequencing was performed at the Emory Genomics Center. We wish to thank Tauqeer Alam at Emory University, for advice on phylogenetic methods, and Sinisa Vidovic and Sidarath Dev from the Vaccine and Infectious Disease Orgarnization- International Vaccine center at the University of Saskatchewan, Canada for their help in antimicrobial susceptibility determination and preliminary genomic assessments. We also thank the Broad Institute for pre-publication release of genomic data used in this study.

Additional Information and Declarations

Competing Interests

Author Contributions

DNA Deposition

Richard F. Selden and Rosemary S. Turingan are employees of NetBio.

Matthew N. Ezewudo conceived and designed the experiments, performed the experiments, analyzed the data, contributed reagents/materials/analysis tools, wrote the paper, prepared figures and/or tables, reviewed drafts of the paper.

Sandeep J. Joseph performed the experiments, analyzed the data, contributed reagents/materials/analysis tools, wrote the paper, prepared figures and/or tables, reviewed drafts of the paper.

Santiago Castillo-Ramirez performed the experiments, analyzed the data, contributed reagents/materials/analysis tools, wrote the paper, reviewed drafts of the paper.

Deborah Dean contributed reagents/materials/analysis tools, prepared figures and/or tables, reviewed drafts of the paper.

Carlos del Rio, Xavier Didelot, Richard F. Selden and Rosemary S. Turingan contributed reagents/materials/analysis tools, reviewed drafts of the paper.

Jo-Anne Dillon performed the experiments, analyzed the data, contributed reagents/materials/analysis tools, reviewed drafts of the paper.

William M. Shafer and Magnus Unemo performed the experiments, analyzed the data, contributed reagents/materials/analysis tools, prepared figures and/or tables, reviewed drafts of the paper.

Timothy D. Read conceived and designed the experiments, contributed reagents/materials/analysis tools, wrote the paper, prepared figures and/or tables, reviewed drafts of the paper.

The following information was supplied regarding the deposition of DNA sequences:

Genomic data can be found at http://www.ncbi.nlm.nih.gov/bioproject/, BioProject numbers: PRJNA209307, PRJNA209312, PRJNA209316, PRJNA209319, PRJNA209320, PRJNA209333, PRJNA209334, PRJNA209340, PRJNA209342, PRJNA209343, PRJNA209345, PRJNA209347, PRJNA209351, PRJNA209352, PRJNA209373, PRJNA209375, PRJNA209376, PRJNA209465, PRJNA209466, PRJNA209468, PRJNA209470, PRJNA209476, PRJNA209477, PRJNA209479, PRJNA209788, PRJNA209483, PRJNA209492, PRJNA209493, PRJNA209497, PRJNA209498, PRJNA209499, PRJNA209501, PRJNA209502, PRJNA209504, PRJNA209507, PRJNA209508, PRJNA209510, PRJNA209512, PRJNA209514, PRJNA209517, PRJNA209522, PRJNA209596, PRJNA209599, PRJNA209616, PRJNA209631, PRJNA209633, PRJNA209635, PRJNA209637, PRJNA209638, PRJNA209639, PRJNA209640, PRJNA209642, PRJNA209647, PRJNA209648, PRJNA209650, PRJNA209654, PRJNA209655, PRJNA209659, PRJNA209310, PRJNA209310, PRJNA209335, PRJNA209641.

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
