# Peer review of "Population structure of Neisseria gonorrhoeae based on whole genome data and its relationship with antibiotic resistance"

_PeerJ, doi:10.7717/peerj.806_

## Round 0.1 · original submission · Minor Revisions

· Academic Editor

Minor Revisions

Dear Matthew Ezewudo, Timothy Read and colleagues,

Thank you for submitting this manuscript to PeerJ.

It appears to represent to date the most comprehensive analysis, based on the yet largest available whole-genome data resource of N. gonorhoeae strains from different geographical regions of the world. Most of the included genomes were de novo sequenced. Despite the fact that N. gonorrhoeae is a common sexually transmitted pathogen which is very prevalent in many regions of the world, its epidemiology, ancient and current global spread, and its possibilities and opportunities of global genomic admixture are not very clear yet.
Hence, the reviewers concur that the manuscript is a valuable resource for N. g. diversity, authors present state-of-the-art genome data analyses. They agree that, although probably not sufficient yet to answer all questions about N. gonorrhoeae worldwide populations, phylogeny and evolutionary traits, this work comprises a lot of valuable data.

Both of the reviewers, and this appears to be their main point of criticism, are concerned about the choice of the sample set with respect to characterizing the whole population structure of N. gonorrhoeae. Strains appear to stem mostly from metropolitan areas, and rural or remote areas have apparently not been sampled. There seems to be geographical as well as temporal bias within the sample set. Since they come to the conclusion that N. gonorrhoeae has no strong geographic separation, which ones are the optimal methods to clarify whether this sample set or the methods are sufficient for drawing this conclusion? Is the sample size large and diverse enough for the analyses performed, and if not, how can this be overcome? One of the reviewers also has some doubts that using dN/dS as a measure for positive or diversifying selection is justified in the collection of strains (homoplasies seem to have been carefully removed for this analysis, and dN/dS was used to determine genetic drift and not positive selection)? As one example, provided by reviewer 1, how can the authors estimate the time separation to the most common ancestor of their strain sample? These questions all merit quite some attention to detail, when using methods, explaining the rationale for the methodology, and discussing the study.

The first reviewer would like the authors to be more precise in general with their phrasing; for instance, notes that the authors speak of “whole genome” data, which might be misleading. Most draft-genomes indeed suffer from multiple sequencing errors, and in particular from a lack of precise contig order. Please specify carefully that the genomes are draft genomes and not single contiguous error-corrected genomes.

At places, the text seems to be very much condensed, which might be a source of misconceptions.

Additional remarks and questions:

Is there any hypothesis to account for the finding that metabolic genes appear to be under positive selection?

Could the conclusions about the spread of antibiotics resistance in the population be made clearer, is this part conclusive? This part appears to remain somewhat unclear - tetracycline R for instance seems to be widely distributed in the whole sample set (Fig. 5) and the other resistance phenotypes seem to be rather sporadic, so is the present conclusion about clade-specific resistance profile really justified? Should more b-lactams be tested in addition to the ones currently in use for treatment?

Strain names in Fig 4 are hard to read

Additional important genome information - number of contigs, approximate genome size - for all de novo-sequenced genomes should be included in table (Table S1).

Please address all major and minor reviewers’ comments, concerning text, methods and figures, and make sure to include a point-to-point letter with responses to all reviewers’ and editor’s comments and questions with your revision.

Best seasons’ greetings, Christine Josenhans

Reviewer 1 ·

Basic reporting

Ezewudo and colleagues analyze the population structure of Neisseria gonorrhoeae by genome-wide sequencing based on a world-wide sample of strains with a particular focus on antibiotic resistance.
The manuscript itself is well written, the figures are carefully prepared and the references appropriately and comprehensively cited. The strucuture of the article also conforms toone of the templates.

Experimental design

Overall, the experimental design and the methods applied are stat-of-the-art, and the results obtained of considerable interest at least to infectious disease specialists and population biologists.

Validity of the findings

The conclusions are mostly supported by the data. However, there seem to me some points that should deserve more attention by the authors prior considering publication of the manuscript.
Specific comments:
1. Title, abstract and elsewhere: The authors repeatedly speak of “whole genome” data and sequencing. This is at least grossly misleading. As the term implies, “whole” genome plainly means whole genomes but not draft assemblies of shotgun sequencing data. A truly complete genome is a single contiguous, i. e. fully assembled and error-corrected sequence and makes the fundamental hereditary content of an organism finite (Fraser et al. 2002; Mardis et al. 2002; Parkhill 2002). Compared to whole genomes draft-genomes suffer from a lack of contig order. In addition, the quality of unfinished partial shotgun (draft) sequences does also often not allow to reliably discerning very small, single base-pair changes from just sequencing errors. In the present case, it would therefore be more appropriate to speak of, e.g., genome-wide sequencing and draft-genomes.
2. Lines 61ff: Can the authors somehow estimate whether their strain sample is representative for the entire gonococcal population and if so how? In particular, are there any other, larger and more comprehensive population studies available that could be used as a kind benchmark data set to compare selected population genetic parameters with?
3. Lines 98ff: The author write “The nucleotide sequences of all the core genes were concatenated together and whole-genome nucleotide alignment was conducted […]”. This is self-contradictory. By concatenating core gene sequences one ends up with the coding part of the core genome which by definition is only a part of an organism’s (whole) genome.
4. Lines 172ff: Amongst others, recombination can also adversely affect the inference of substitution rates (Posada et al. 2002). How did the authors separate the effect of recombination from point mutation in their substitution rate estimation? And have they tested for possible substitution rate heterogeneity among the different subgrups
5. Lines 179 ff and elsewhere: As also cited by the authors, Rocha et al. (Rocha et al. 2006) as well as Kryazhimsky and Plotkin (Kryazhimskiy & Plotkin 2008) have demonstrated that dN/dS is inappropriate as a measure of positive selection when comparing very closely related species or even strain from the same species, within which dN/dS can be inflated by segregating nonsynonymous polymorphism. In fact, the hallmark signature of positive selection over divergent lineages, dN/dS >1, is violated within a population (Kryazhimskiy & Plotkin 2008). What is then the justification for using this method on the given strain sample? For example, is the estimated time to the most recent common ancestor of the strains in their sample expected to be large enough to allow for effective positive selection to be detected?

References
Fraser CM, Eisen JA, Nelson KE, Paulsen IT, and Salzberg SL. 2002. The Value of Complete Microbial Genome Sequencing (You Get What You Pay For). J Bacteriol 184:6403-6405.
Kryazhimskiy S, and Plotkin JB. 2008. The population genetics of dN/dS. PLoS Genet 4:e1000304.
Mardis E, McPherson J, Martienssen R, Wilson RK, and McCombie WR. 2002. What is finished, and why does it matter. Genome Res 12:669-671.
Parkhill J. 2002. The importance of complete genome sequences. Trends Microbiol 10:219-220.
Posada D, Crandall KA, and Holmes EC. 2002. Recombination in evolutionary genomics. Annu Rev Genet 36:75-97.
Rocha EP, Smith JM, Hurst LD, Holden MT, Cooper JE, Smith NH, and Feil EJ. 2006. Comparisons of dN/dS are time dependent for closely related bacterial genomes. J Theor Biol 239:226-235.

Additional comments

No comments.

Reviewer 2 ·

Basic reporting

The text of the manuscript is clear and well written. Many of the figures, however, could be improved.

Figure 1. Need to improve the visualization of the phylogeny – it’s unclear what the thick lines represent – they could also be less thick to facilitate visualization. Are the grey dots supposed to indicate those branches with bootstrap values >80%? How does BAPS group correlate with MLST?

Figure 2. Please clarify the font and orientation of the labels to make it easier to read the figure. Please also clarify how this figure is supposed to be read. What does 0 represent for each of the groups? How is the information in Figure 2 different from that in Figure 3?

Figure 3. Unclear how to read this figure. What is on the y-axis?

Figure S1. Need to correct legend so that it reads “N. gonorrhoeae”. If N=81, why is the “red line” designating core genes set at around 75, and why are there no genes that appear in all 81? Is the x-axis labeling incorrect, or is there something else going on?

Figure S3. How is this figure generated? What is the X-axis?

For table S1, instead of simply listing presence and absence of mutations, could the authors please report the actual mutations?

Experimental design

The article by Ezewudo et al. analyzes a set of genomes of Neisseria gonorrhoeae with the goals of describing the population structure of a more global set of strains than has been studied to date. Overall, this is a well-done study, and a solid contribution to the literature. I have a number of questions and points, detailed below.

How were the isolates for this study selected? This seems a convenience sample. Is that correct? Or were these particular isolates selected on the basis of some characteristic? If so, why were multiple isolates of the same MLST selected? If one of the goals was geographic diversity, then how was the appropriate geographic distribution decided on?

On a related note, I agree that “there is a need for studies aimed at understanding the global N. gonorrhoeae population structure.” But the relationship between geographic distance and phylogenetic distance for any given pair of gonococcal isolates isn’t actually clear to me, though it seems implicit in the idea that selection of a “global” collection of isolates will be more reflective of gonococcal diversity (e.g., lines 45-52). As gonococci seem to spread around the world relatively quickly, I wonder to what extent the gonococci circulating in a metropolitan city may reflect much of the world’s diversity (even if not in proportion to global prevalence).

In the methods, it seemed like the bioinformatics tools were being reported in the order in which they were used – if that is the case, I wonder how homoplasies were identified prior to phylogenetic inference.

For the concatenation of the core genes, to what extent was synteny maintained? If it wasn’t, what’s the impact on the output of ClonalFrame?

For determination of gene flow among the Neisseria genus, which 14 other Neisseria genus genomes were used? Please provide the accession numbers. Please also clarify the sentence beginning “We also filtered off hits…” It’s not clear what was done here.
Lines 219-221 – what does “evenly distributed across the different clades” mean? I’m taking issue with the description of “evenly” because it’s not actually clear that it is even. Is there a way of quantifying the distribution?

Please clarify lines 290-292.

For isolates that in which genotype and AMR phenotype were discordant or for which no clear genetic basis for resistance was identified, were MICs double-checked? As there can be variation / error in MICs, it might be useful to confirm that they are accurate.

Validity of the findings

The findings appear congruent with past studies and extend them to help provide an overview of the population structure of N. gonorrhoeae and the indications of geographic differences in selective pressures and strain differences in recombination.

---

## Round 0.2 · accepted · Accept

· Academic Editor

Accept

Dear Dr. Ezewudo,

Thank you for submitting your revised manuscript. All the reviewers' and editorial comments have been appropriately and thoroughly addressed.

Best regards,
Christine Josenhans